# Diffusion MRI reveals in vivo and non-invasively changes in astrocyte function induced by an aquaporin-4 inhibitor

**Clement Debaker**, **Boucif Djemai, Luisa Ciobanu, Tomokazu Tsurugizawa\*, Denis Le Bihan\***

NeuroSpin, Gif-sur-Yvette, France

\* denis.lebihan@gmail.com (DLB); tsurugizawa@gmail.com (TT)

**Data Availability Statement:** All relevant data are within the paper.

**Funding:** This research was supported by a public grant of the French National Research Agency

## Abstract

The Glymphatic System (GS) has been proposed as a mechanism to clear brain tissue from waste. Its dysfunction might lead to several brain pathologies, including the Alzheimer's disease. A key component of the GS and brain tissue water circulation is the astrocyte which is regulated by acquaporin-4 (AQP4), a membrane-bound water channel on the astrocytic end-feet. Here we investigated the potential of diffusion MRI to monitor astrocyte activity in a mouse brain model through the inhibition of AQP4 channels with TGN-020. Upon TGN-020 injection, we observed a significant decrease in the Sindex, a diffusion marker of tissue microstructure, and a significant increase of the water diffusion coefficient (sADC) in cerebral cortex and hippocampus compared to saline injection. These results indicate the suitability of diffusion MRI to monitor astrocytic activity in vivo and non-invasively.

## Introduction

Proper neuronal function necessitates a highly regulated extracellular environment in the brain. Accumulation of interstitial solutes, such as amyloid β and toxic compounds [1] may lead to degenerative diseases, such as Alzheimer's disease [2,3] or even autism [4]. Although the Blood Brain Barrier is thought to be the primary mechanism involved in controlling the brain blood-exchange, the existence of a fluid driven transport system (so-called glymphatic system) via cerebrospinal fluid (CSF) or interstitial fluid (ISF) has been proposed recently as a waste clearance system through the perivascular and interstitial spaces in the brain [1,5]. Hypothetically, CSF crosses the astrocyte end-feet bound to arteries in the perivascular space [6,7]. After washing the interstitial space, the resulting ISF is flushed back outside the brain via veins in the perivascular space. Several factors play a crucial role in the modulation of this clearance system activity, notably sleep and anesthesia, perhaps via a modulation of brain blood volume and pressure [8–11]. This scheme gives astrocytes a crucial role in controlling water movements between the blood and the brain, through a mechanism dependent on Aquaporin-4 (AQP4), a membrane-bound water channel expressed at their end-feet [12,13]. AQP4 is involved in the rapid volume regulation of astrocytes [14] and deletion of the AQP4 gene suppresses the clearance of soluble amyloid β [1].

(project "MrGLY", reference: ANR-17-CE37-0010, DLB=PI). The funders had no role in study design, data collection and analysis, decision to publish, or preparation of the manuscript.

**Competing interests:** The authors have declared that no competing interests exist.

Although most studies have been performed *in vitro*, some studies have shown astrocyte volume changes *in vivo* using 2-photon microscopy. Acute osmotic and ischemic stress induce astrocyte volume changes *in vivo* mice [15], and Thrane et al showed that the astrocyte volume change induced by osmotic stimulation was inhibited in AQP4 KO mice [16]. Overall, such studies suggest that dynamic volume change of astrocytes, through water flux mediated by AQO-4 channels may be associated with CSF flow regulation. Actually, astrocytes end-feet are involved in the CSF/ISF exchanges in perivascular space during sleep/awake cycle [17].

Recently MRI has been proposed as a more versatile approach to investigate the glymphatic system in vivo, using intrathecal or intravenous injections of gadolinium-based contrast agents (GBCAs) as tracers [7,18]. However, this approach remains invasive and, paradoxically, gadolinium has been shown to deposit in the brain [19,20] possibly in relation to a lack of brain drainage [10]. Therefore, alternative methods are needed to investigate the glymphatic system non-invasively, especially in the human brain. Fluid-dynamics driven and BOLD fast MRI have the potential to evaluate CSF pulsations in the ventricles and hemodynamics [11,21,22], while IVIM and diffusion MRI have been shown as promising methods for the evaluation of the ISF [7,23–27].

Those considerations led us to investigate whether diffusion MRI was sensitive to astrocyte activity and, in turn, could become a marker of the overall glymphatic system. Diffusion MRI is exquisitely sensitive to changes in tissue microstructure, notably cell swelling [28]. Diffusion MRI is, for instance, sensitive to astrocyte swelling induced in rodents [29]. Hence, we hypothesized that dynamic changes in astrocytes activity and related volume changes could be monitored directly and non-invasively with diffusion MRI. To test this hypothesis we monitored variations of new diffusion MRI markers, namely the Sindex and the sADC, which have been tailored to increase sensitivity to tissue microstructure through water diffusion hindrance [30,31] upon acute inhibition of astrocyte AQP4 channels in a mouse brain model using 2-(nicotinamide)-1,3,4-thia-diazole (TGN-020), a compound that blocks AQP4 channels *in vivo* in the mouse brain [32].

## Material and methods

### Animal preparation

Thirty-two male C57BL6 mice (16–28 g, 4–10 weeks, Charles River, France) were allocated to two groups. The choice of a mouse brain model was motivated by prospect of using our protocol later to AQP-4 knock-out mice. First, for the TGN-020 group, 16 mice received an intraperitoneal injection of 250mg/kg TGN-020 diluted in a gamma-cyclodextrine solution (10 mM) in order to increase its solubility. Second, for the control group, sixteen mice received an intra-peritoneal injection of the vehicle solution only (10 mM gamma-cyclodextrine in saline). The mice were housed on a 12-hour light-dark cycle and fed standard food ad libitum. Anesthesia was induced using 3% isoflurane in a mix of air and oxygen (air: 2 L/min, O2: 0.5 L/min). Then, 0.015 mg/kg of dexmedetomidine was administered intraperitoneally and followed by a continuous infusion of 0.015 mg/kg/h via subcutaneous catheter and maintained isoflurane at 0.8%.

Throughout the acquisition, the animals' body temperature was maintained between 36.5 and 37.0 ˚C using heated water (Grant TC120, Grant Instruments, Shepreth, UK). To avoid motion-related artifacts the head was immobilized using a bite bar and ear pins. The respiration rate was monitored and stable (60–90 /min) throughout the experiment.

All animal procedures used in the present study were approved by an institutional Ethic Committee (Comité d'Ethique en Expérimentation Animale, Commissariat à l'Energie Atomique et aux Énergies Alternatives, Direction des Sciences du Vivant (Fontenay-aux-Roses,

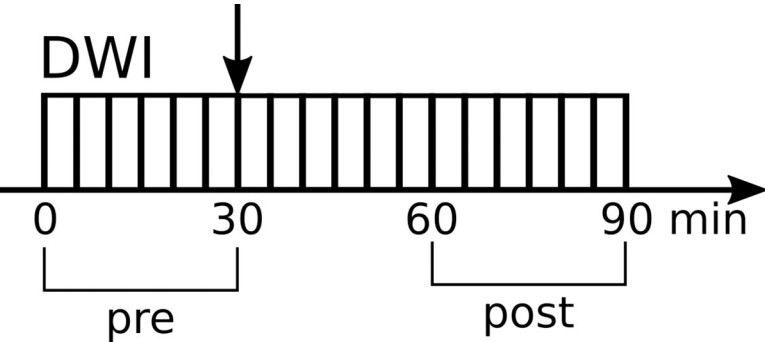

**Fig 1. Schematic figure of MRI experiment protocol.** The vertical arrow indicates the saline or TGN-020 injection. DWI, diffusion weighted MRI; NR, number of repetitions. Horizontal arrow indicates the timeline.

France)) and by Ministère de l'Education Nationale, de l'Enseignement Supérieur et de la Recherche (France) (reference APAFIS#8462-2017010915542122v2) and were conducted in strict accordance with the recommendations and guidelines of the European Union (Directive 2010/63/EU). This manuscript is in compliance with the ARRIVE guidelines (Animal Research: Reporting in Vivo Experiments) on how to REPORT animal experiments.

## MRI experiments

The MRI experiments were conducted on a Bruker 11.7T scanner (Bruker BioSpin, Ettlingen, Germany) equipped with a gradient system allowing a maximum gradient strength of 760 mT/m. A cryo-cooled mouse brain coil was used. Animal positioning was performed using multi-slice fast low angle shot imaging (FLASH, TE/TR = 2.3/120 ms). A global first and second order shim was achieved followed by a local second order shim over the brain parenchyma. Structural (anatomical) images were acquired with the following parameters: T2 TurboRARE sequence, TE/TR = 11.15/2500 ms. Diffusion-weighted echo-planar imaging (DW-EPI) data sets were acquired with the following parameters: 150x150x250 $\mu m^3$ resolution, 3 b-values (0, 250, 1750) s/mm$^2$ along 6 directions, NA = 4, TE/TR = 36.3/2300 ms, 18 slices, diffusion time = 24 ms, total scan time = 5 min. The area covered by the 18 slices of the scans encompassed 5 mm in the axial plane with the center of the slab positioned at the middle of the brain. Six DW-EPI sets were first acquired (baseline) before TGN-020 or vehicle was injected. Then, after the injection, 12 additional DW-EPI data sets were acquired (Fig 1) every 5 minutes.

## Data processing

**Preprocessing.** Regions of interest (ROIs) were originally created using the Allen Brain Atlas [33] and registered to a mouse brain MRI template to create a labeled atlas. We referred to previous studies based on microscopy [34] to assess the vascular distribution within the selected brain structures. The atlas was then co-registered with the b = 0 images of the first scans for each subject, and not the other way around to avoid changing raw voxel signals. Those preprocessing steps, also including radiofrequency bias field correction [35] and denoising were performed using ANTs (https://stnava.github.io/ANTs/) [36] (Fig 2). We assumed geometric distortion to be negligible; this condition was qualitatively checked on several mice for all b values. Furthermore, the standard-deviation of the signal intensity across scans was evaluated for each b value and each voxel (after Gaussian smoothing) was estimated as a marker of motion or instability. A threshold of 4% was used to flag unstable voxels. Data sets were at least one of the ROIs contained flagged voxels were discarded.[37].

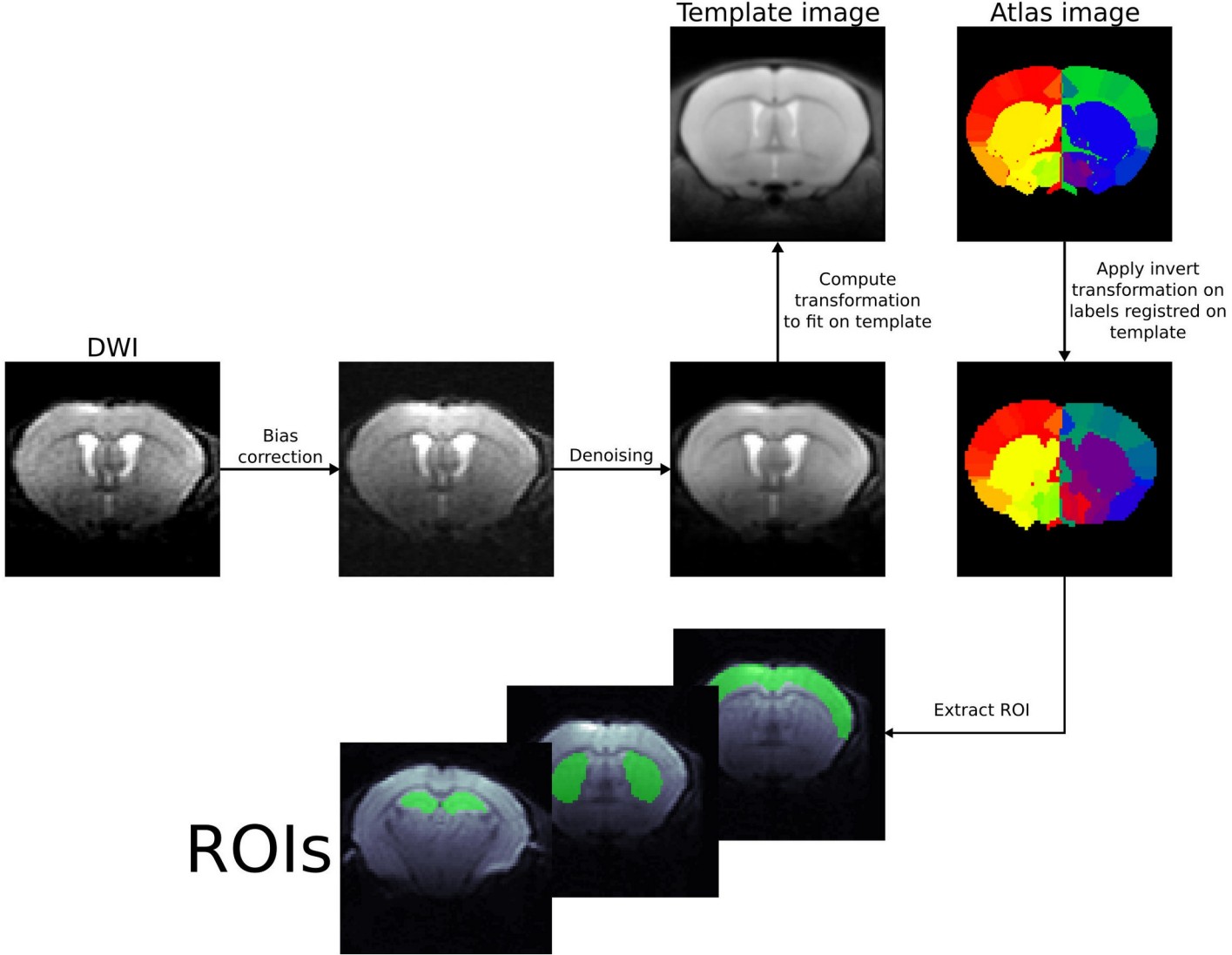

**Fig 2. Example of ROI registration with ANTs on a mouse brain.**

**DWI analysis.** First, a shifted ADC (sADC) was computed using signal acquired at two key b values (instead of the standard values of 0 and 1000s/mm$^2$), chosen to optimize signal sensitivity to both Gaussian and non-Gaussian diffusion which is more sensitive to tissue microstructure through hindrance effects [30,31].

$$sADC = \frac{\ln\left(\frac{S_{Lb}}{S_{Hb}}\right)}{Hb - Lb} \tag{1}$$

where Lb is the low key b value (250s/mm$^2$), Hb the high key b value (1750 s/mm$^2$), $S_{Lb}$ the signal at low key b value and $S_{Hb}$ the signal at high key b value.

Second, data were analyzed using the Sindex method [31]. The Sindex diffusion marker has been designed to identify tissue types or conditions based on their microstructure [31]. The Sindex was calculated from the direction-averaged, normalized signals, $S_V(b)$ in each voxel, as the algebraic relative distance between the vector made of these signals and those of 2 signature

tissue signals $S_A$ in condition A, and $S_B$ in condition B, at each key b value as [31]:

$$SI(V) = \left\{ \max\left( \frac{[dS_V(Hb) - dS_V(Lb)]}{[dS_B(Hb) - dS_B(Lb)]}, 0 \right) - \left[ \max\left( \frac{[dS_V(Hb) - dS_V(Lb)]}{[dS_A(Hb) - dS_A(Lb)]}, 0 \right) \right] \right\} (2)$$

with $dS_{V,A,B}(b) = [S_{V,A,B}(b) - S_N(b)]/S_N(b)$. $S_N$ is taken as an intermediate signal between $S_A$ and $S_B$. SI was then linearly scaled as *Sindex* = (SI+1)*25+25 to be centered at 50. A tissue with status similar to condition A has *Sindex* = 75, while for a status similar to condition B one has *Sindex* = 25. The library of the 2 reference diffusion MRI signals was built in advance from previously acquired data, one representing a generic mouse brain tissue (B) and another derived by simulating a moderate increase in diffusion hindrance (A) using the Kurtosis diffusion model [31]. For this study, $S_A(Lb) = 0.858$; $S_A(Hb) = 0.370$; $S_B(Lb) = 0.855$; $S_B(Hb) = 0.317$. Obviously the Sindex is expected to vary widely across brain regions according to their degree of diffusion hindrance (Fig 3), for instance white matter regions have higher Sindex values than gray matter regions. However, our focus was on the local changes in Sindex values induced by the injection of TGN-020, reflecting local changes in the degree of diffusion hindrance (a decrease in Sindex reflecting a decrease in hindrance). Sindex and sADC were calculated on a voxel-by-voxel basis to generate parametric maps. Calculation was also performed on a ROI level, averaging signals from all voxels within the ROI. ROIs were placed over the cerebral cortex, the striatum and the hippocampus (whole hippocampus, CA3 region and dentate gyrus, DG). The hippocampus is known to be reached in AQP4 expression [38,39].

Values obtained before (6 first scans) and after injection (6 last scans) were averaged in each ROI for each animal. The parameter time course was also calculated by averaging 3 successively acquired datasets (resulting in six time points with a resolution of 15 minutes) for each animal. Before averaging across individual subjects, we performed an outlier exclusion using a z-score filter (z>3) calculated over the subject's parameter values for each time point.

## Statistical analysis

The statistical tests were performed in python (Python Software Foundation. Python Language Reference, version 3.7. Available at http://www.python.org). We performed a paired two sample t-test between pre and post-injection data in each group and a pair-wise t-test with a post-hoc correction for multiple comparison to compare the post-injection data between two groups and the pre-injection data between two groups.

For the time course data, we performed a two sample t-test between the two groups for each time point.

## Results

### Averaged Sindex and sADC following vehicle or TGN-020 injection

Fig 3 shows brain maps of Sindex and sADC for a representative mouse following vehicle and TGN-020 injection. Differences between the two conditions are readily visible with a decrease in Sindex and an increase in sADC following TGN-20 injection. Those changes are quantitatively assessed in Fig 4. Fig 4A–4C show the Sindex averaged over six scans before and after the injection in different locations, i.e., cerebral cortex, hippocampus and striatum, in vehicle and the TGN-020 group. TGN-020 injection resulted in significant decrease in Sindex in the cortex (p = 0.0061) and the hippocampus (p = 0.00069) but not in the striatum (p = 0.26). No significant change of Sindex was observed following vehicle injection in those locations (p = 0.94 in cortex, p = 0.85 in hippocampus and p = 0.14 in striatum). Fig 4D–4F show the sADC averaged over six scans before and after the injection in the same locations as for the

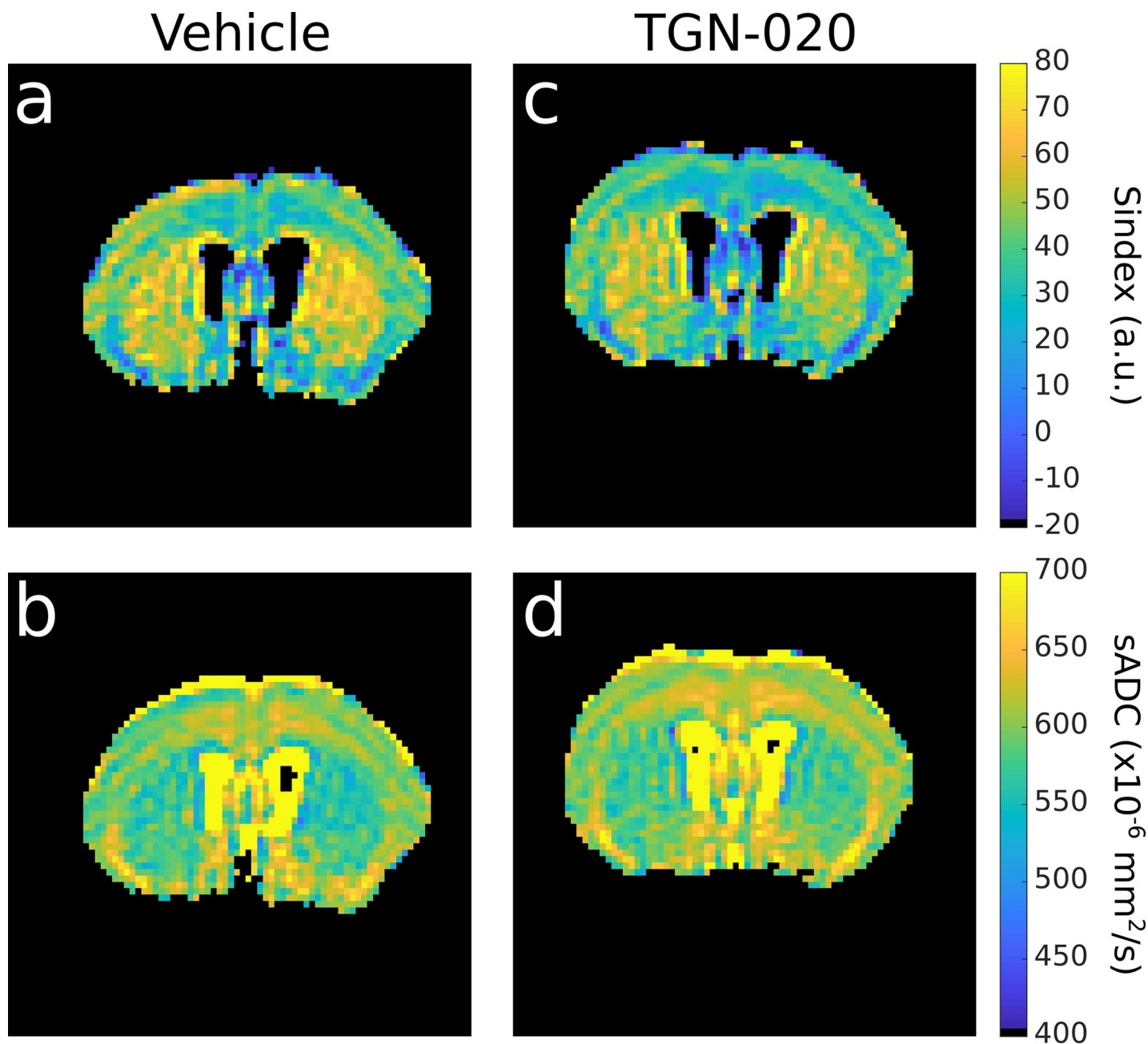

**Fig 3. Sindex (first row) maps and sADC (second row) maps for a representative mouse from vehicle group (first column) and from TGN-020 group (right column).** These maps were obtained by averaging the 6 last time points.

Sindex, in vehicle and TGN-020 group. TGN-020 injection resulted in significant increase in sADC in the cortex (p = 0.0064) and the hippocampus (p = 0.00068) but not in the striatum (p = 0.26). No significant change of sADC was observed following vehicle injection in any of the ROIs considered (p = 0.91 in cortex, 0.78 in hippocampus and 0.14 in striatum). We also investigated subregions of the hippocampus CA3 and DG (S1 Fig). The TGN-020 injection significantly decreased Sindex and increased sADC values both in the DG and the CA3 (Sindex in DG: -11.8, p = 4.8e-7 and in CA3: -4.8, p = 0.00012; sADC in CA3: +10.2, p = 0.00013,

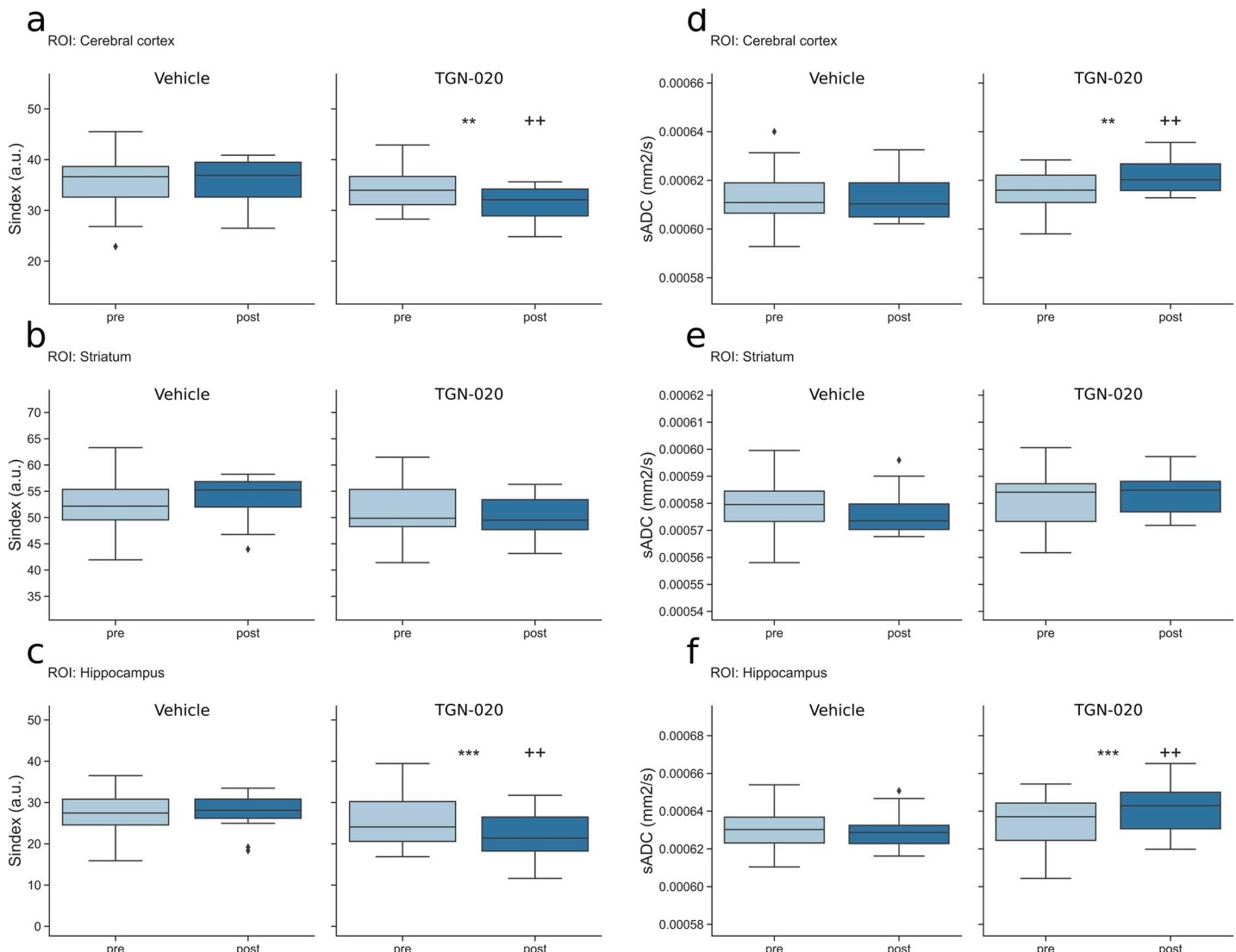

**Fig 4.** Boxplot of Sindex and sADC in the cortex (a: Sindex, d: sADC), the striatum (b: Sindex, e: sADC) and hippocampus (c: Sindex, f: sADC). Left, Vehicle group; right TGN-020 group in each figure. Pre, pre-injection (light blue); post, post-injection (dark blue) of vehicle or TGN-020 group. *: p<0.05, **: p<0.01, ***: p<0.001 are the result for the paired t-test between pre and post-injection for each group. +: p<0.05, ++: p<0.01, +++: p<0.001 are the result for a t-test with posthoc correction for multiple comparison between post-injection of the two groups. Diamonds represent outliers, a point is defined as an outlier if its value is below Q1–1.5×IQR or above Q3 + 1.5×IQR, where Q1 is the first quartile, Q3 the third quartile and IQR the interquartile range.

sADC in DG: +22.5, p = 7.14e-7). However, the baseline for Sindex and sADC in DG was found noisy and not stable (S1B and S1F Fig) due to the smaller size of DG resulting in a very low voxel count compared to the other ROIs (voxel count for DG, CA3, hippocampus, striatum, cortex were 85, 181, 763, 2069 and 8270, respectively).

## Sindex and sADC time courses following vehicle or TGN-020 injection

We then investigated the time course of the Sindex and sADC for each group. The same ROIs were used as for the averaged Sindex and sADC values. Fig 5A–5C represents the Sindex time course for each ROI. A decrease of Sindex was observed after the injection of TGN-020 but not with vehicle injection in all ROIs. The Sindex significantly decreased (around 9% in the

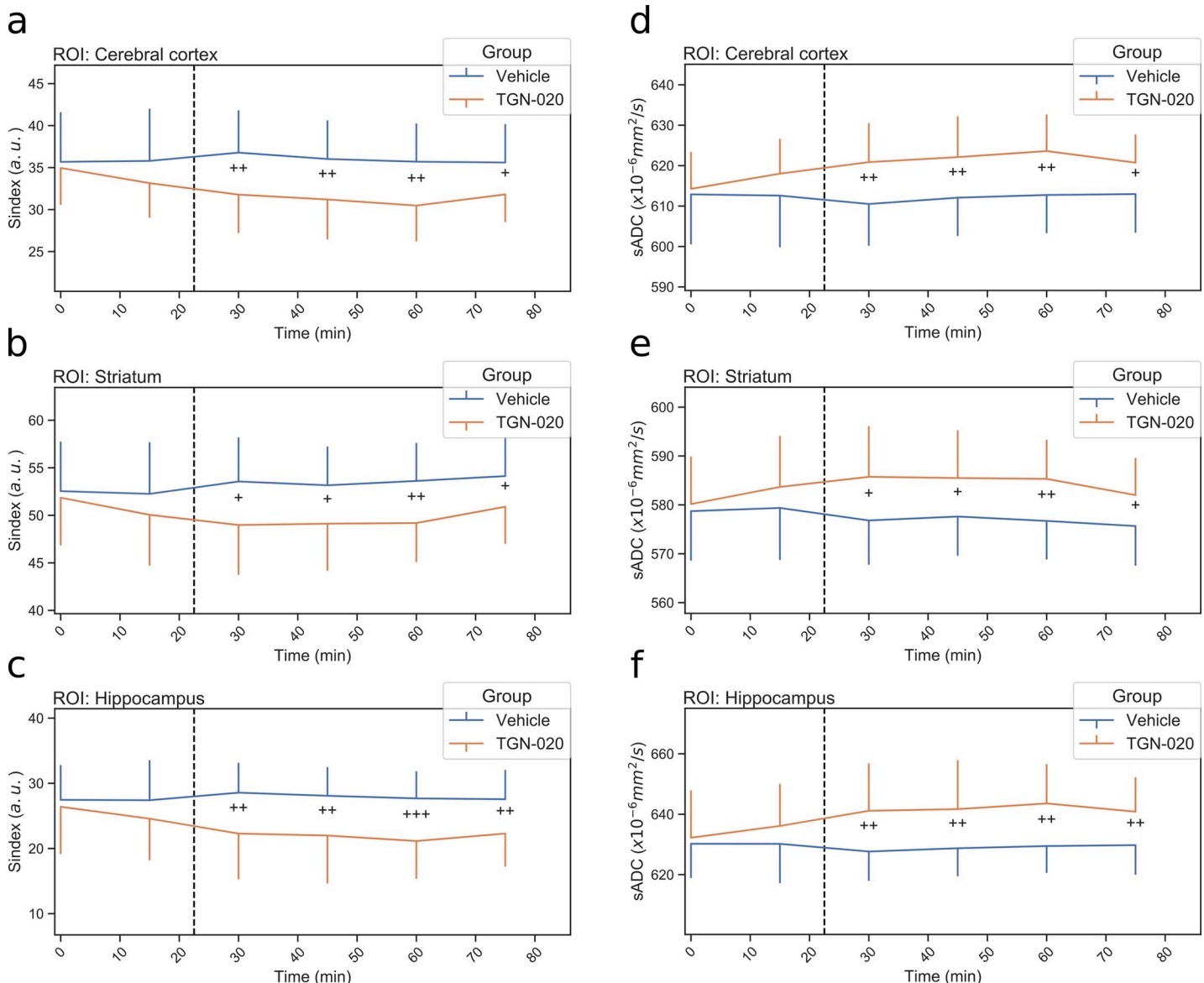

**Fig 5.** Time course of Sindex and sADC in the cortex (a: Sindex, d: sADC), the striatum (b: Sindex, e: sADC) and hippocampus (c: Sindex, f: sADC). Blue line is vehicle group and orange line is TGN-020 group. Error bar shows standard deviation. The dashed line represents the injection time. +: $p<0.05$, ++: $p<0.01$, +++: $p<0.001$ by two sample t-test between the two groups for each time point.

cortex) following TGN-020 injection in all ROIs and continued until the end of the scanning period. Fig 5D–5F show the sADC time course with an opposite trend compared to Sindex in all ROIs.

## Discussion

Several studies have underlined the potential of Diffusion MRI to investigate the glymphatic system [7,23–25]. Astrocytes are considered to play a key role in brain waste clearance through membrane AQP4 channels expressed at their end-feet [12]. The ADC has been proposed as a biomarker of AQP gene expression, as earlier studies have demonstrated that the ADC obtained with high b values was correlated with the amount of increasing expression of AQP1

in glioblastoma cell lines [40]. In this study, we have investigated whether two diffusion MRI markers more sensitive to tissue microstructure than the ADC, namely the Sindex and sADC, could reveal changes in astrocyte activity induced by TGN-020.

Following AQP4 channel inhibition with a TGN-020 solution a decrease in Sindex and increase in sADC were readily observed in the cortex, more in the hippocampus, but not in the striatum, reflecting local differences in astrocyte [41] and vascular density [34]. In the hippocampus changes in Sindex and sADC were larger in DG than in CA3 layer, presumably in line with differences in AQP4 channels expression on astrocytes [39]. However, further validation is required as the baseline Sindex and sADC values were not stable due to the small count in the DG ROI (see Result). Histological studies would also confirm differences in astrocyte density and volume. Nonetheless, those results suggest that diffusion MRI is sensitive to astrocyte activity and, indirectly, to the status of the glymphatic system. Those diffusion MRI markers provide a higher sensitivity to small changes in tissue features by encompassing in a single marker Gaussian and non-Gaussian diffusion effects. Furthermore they are easy to calculate and are not diffusion signal model dependent, such as the kurtosis model [31]. The low key b value used in this study was high enough (250 s/mm$^2$) to make perfusion-related IVIM effects negligible. Hence the observed changes in the diffusion markers reflect genuine tissue microstructure related diffusion effects and not perfusion effects (cerebral blood flow) which have already been reported observed with TGN-020 [42]. Indeed, acquisition of perfusion-driven IVIM data could be valuable to get information on the vascular distribution which is known to vary across brain structures [34], but this was not possible within our protocol timeframe to guaranty animal stability.

The Sindex decrease and the sADC increase jointly point out to a decrease in the amount of hindrance for water diffusion in astrocyte rich areas (cortex and hippocampus) under acute AQP4 channel inhibition induced by TGN-20. Based on established diffusion MRI mechanisms [28] this hindrance decrease suggests an astrocyte volume reduction [43,44] associated with an increase of the ISF (were diffusion is tortuous) [45] or an increase in astrocyte membrane permeability and water exchange. Indeed, astrocytes rapidly regulate their volume throughout AQP4 channels. Those results obtained by acute AQP4 inhibition contrast earlier reports using chronic models, such as an ADC decrease observed during AQP4 inhibition with interfering RNA [46] or change in ADC in AQP4 knockout mice [47]. Beside the higher sensitivity to tissue microstructure of the sADC over the ADC, such discrepancy could possibly result from the modified astrocyte phenotype associated with a long-term inhibition of AQP4 expression found in those previous studies [48]. The sub-acute or chronic inhibition of AQP4 activity by AQP4-antibodies or small interfering RNA duplexes alter astrocyte morphology and decrease water permeability [49,50] which could result in an ADC decrease. Also, the effects we observed in baseline conditions should be distinguished from those obtained in conditions of neuronal activation for which AQP4 inhibition by extracellular acidification results in astrocyte swelling, capillary lumen expansion and Virchow-Robin space reduction [51]. Further studies should aim at precising the mechanisms involved in the observed effects, however histology would require the brain to be fixed, preventing astrocyte volume changes to be monitored before and after TGN-020/saline injection within the same animal. In vivo fluorescent imaging could be used, but only in the cortex. Clearly, the detailed mechanisms underlying acute AQP4 channel inhibition by TGN-20 and chronic AQP4 knockout mice models are lacking. Diffusion MRI has the potential to clarify those mechanisms, notably through non-Gaussian diffusion markers, such as the Sindex and the sADC. Fractional anisotropy (FA) measurements results were not reported as they are relevant only to areas exhibiting diffusion anisotropy, ie, white matter given the spatial resolution of our images. Beside, due to noise FA

values are highly corrupted and often reflect mainly variations in underlying ADC (here sADC) [52].

Another issue to consider is that our studies were obviously performed under anesthesia. Anesthetic drugs are known to impact intracranial pressure, which could interfere with CSF-ISF exchanges. For instance, dexmedetomidine and ketamine enhance CSF influx along-side perivascular spaces [53]. Moreover, similar to what happens during sleep, anesthesia is associated with a substantial increase in both perivascular and extracellular space volume [54]. Thus, it would be interesting to check whether the effects we have observed with TGN-20 per-sist in awake animals. A recent study has shown that, while the baseline ADC did not change between anesthetic and awake conditions, DOTA-Gd accumulation was significantly sup-pressed during anesthesia in contrast enhanced MRI [55].

## Conclusion

Work remains to better understand the mechanisms involved in brain waste clearance through a so-called glymphatic system and the contribution of astrocytes to this system. The reference method to investigate the glymphatic system in preclinical settings relies on the intracisternal injection of gadolinium in the cisterna magna [56]. Our results show that changes in astrocyte activity thought to regulate CSF-ISF exchanges, an important component of glymphatic sys-tem functionality, could be monitored non-invasively with diffusion MRI, especially through its Sindex metric. Further studies will be required to more directly establish the potential of diffusion MRI to monitor the glymphatic system. Due to its complete non-invasiveness, this new approach could be used for clinical studies to confirm or infirm the existence of such a glymphatic system in humans [24].

## Supporting information

**S1 Fig.** Boxplot and time course of Sindex in the DG (a: boxplot, b: time course) and in the CA3 (c: boxplot, d: time course). Boxplot and time course of sADC in the DG (e: boxplot, f: time course) and in the CA3 (g: boxplot, h: time course). Left, Vehicle group; right TGN-020 group in each figure. Pre, pre-injection (light blue); post, post-injection (dark blue) of vehicle or TGN-020 group in boxplot. *: $p < 0.05$, **: $p < 0.01$, ***: $p < 0.001$ are the result for the paired t-test between pre and post-injection for each group. +: $p < 0.05$, ++: $p < 0.01$, +++: $p < 0.001$ are the result for a t-test with posthoc correction for multiple comparison between post-injection of the two groups. Diamonds represent outliers, a point is defined as an outlier if its value is below Q1–1.5×IQR or above Q3 + 1.5×IQR, where Q1 is the first quartile, Q3 the third quartile and IQR the interquartile range. Blue line is vehicle group and orange line is TGN-020 group. Error bar shows standard deviation. The dashed line represents the injection time. +: $p < 0.05$, ++: $p < 0.01$, +++: $p < 0.001$ by two sample t-test between the two groups for each time point. Note: The voxel count in DG was very small (85 versus 181 in CA3) resulting in very noisy data. The apparent significant difference in sADC and Sindex values between the TGN-020 and saline groups one point before baseline (b, f) might result from an underestimation of the standard-deviation. The difference becomes largely more significant after injections, overcom-ing any uncertainty in standard-deviation estimates.
(TIF)

## Author Contributions

**Conceptualization:** Tomokazu Tsurugizawa, Denis Le Bihan.

**Data curation:** Clement Debaker, Boucif Djemai, Tomokazu Tsurugizawa.

**Formal analysis:** Clement Debaker.

**Funding acquisition:** Tomokazu Tsurugizawa, Denis Le Bihan.

**Investigation:** Luisa Ciobanu, Tomokazu Tsurugizawa, Denis Le Bihan.

**Methodology:** Luisa Ciobanu, Tomokazu Tsurugizawa, Denis Le Bihan.

**Software:** Denis Le Bihan.

**Supervision:** Tomokazu Tsurugizawa, Denis Le Bihan.

**Validation:** Tomokazu Tsurugizawa, Denis Le Bihan.

**Visualization:** Clement Debaker.

**Writing – original draft:** Clement Debaker, Tomokazu Tsurugizawa, Denis Le Bihan.

**Writing – review & editing:** Clement Debaker, Boucif Djemai, Luisa Ciobanu, Tomokazu Tsurugizawa, Denis Le Bihan.

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
