## [Decision Letter · Decision Letter 0]

3 Mar 2020

PONE-D-20-03675

Diffusion MRI reveals in vivo and non-invasively changes in astrocyte function induced by an aquaporin-4 inhibitor.

PLOS ONE

Dear Pr. Le Bihan,

Thank you for submitting your manuscript to PLOS ONE. After careful consideration, we feel that it has merit but does not fully meet PLOS ONE’s publication criteria as it currently stands. Therefore, we invite you to submit a revised version of the manuscript that addresses the points raised during the review process.

How do you know that the observed diffusion changes are related to the glymphatic system?How do you explain the DG change, given the mentioned AQP4 abundant?Do you have the data to identify regions with high vascular presence to look at the Si and sADC changes on those area?In addition to sindex, the authors could use FA for comparison.Response to reviewer’s questions.

We would appreciate receiving your revised manuscript by Apr 17 2020 11:59PM. To enhance the reproducibility of your results, we recommend that if applicable you deposit your laboratory protocols in protocols.io, where a protocol can be assigned its own identifier (DOI) such that it can be cited independently in the future. For instructions see: http://journals.plos.org/plosone/s/submission-guidelines#loc-laboratory-protocols

We look forward to receiving your revised manuscript.

Kind regards,

Quan Jiang, Ph,D.

Academic Editor

PLOS ONE

Journal Requirements:

"All animal procedures used in the present study were approved by an institutional Ethic Committee and government regulatory agency (reference APAFIS#8462-2017010915542122v2) and were conducted in strict accordance with the recommendations and guidelines of the European Union (Directive 2010/63/EU). This manuscript is in compliance with the ARRIVE guidelines (Animal Research: Reporting in Vivo Experiments) on how to REPORT animal experiments.".

i) Please amend your current ethics statement to include the full name of the ethics committee that approved your specific study.

ii) Once you have amended this/these statement(s) in the Methods section of the manuscript, please add the same text to the “Ethics Statement” field of the submission form (via “Edit Submission”).

For additional information about PLOS ONE submissions requirements for ethics oversight of animal work, please refer to http://journals.plos.org/plosone/s/submission-guidelines#loc-animal-research

Reviewers' comments:

Reviewer's Responses to Questions

**Comments to the Author**

1. Is the manuscript technically sound, and do the data support the conclusions?

Reviewer #1: Yes

Reviewer #2: Yes

2. Has the statistical analysis been performed appropriately and rigorously? 

Reviewer #1: Yes

Reviewer #2: Yes

3. Have the authors made all data underlying the findings in their manuscript fully available?

Reviewer #1: No

Reviewer #2: Yes

4. Is the manuscript presented in an intelligible fashion and written in standard English?

Reviewer #1: Yes

Reviewer #2: Yes

5. Review Comments to the Author

Reviewer #1: Authors presented a very interesting study, evaluating the utility of diffusion MRI to map glymphatic function, through the modification of the AQP4 expression. This is of high scientific and clinical value, very timely and of high impact.

The experiment is well designed, and the execution is sound. This reviewer has below comments regarding the validity and conclusions. Figures quality/dpi are low!

Major comments:

1. The main limitation is interpreting diffusion measures as glymphatic system function. AQP4 is also involved in brain inflammation and regulation of extracellular space volume. How do you know that the observed diffusion changes are related to the glymphatic system?

Even though authors were cautious about the interpretation of their results, they should probably discuss other potential explanations for their findings and adjust their main conclusion accordingly.

2. Authors reported significant differences in both CA3 and DG of the TGN-020 group. How do you explain the DG change, given the mentioned AQP4 abundant?

3. Regarding above comment: Do you have the data (for example SWI or high resolution T2) to identify regions with high vascular presence to look at the S_i and sADC changes on those area? That could potentially help identify the underlying AQP4 involvement.

4. Based on ref [36]: “AQP4 protein levels were highest in the cerebellum with lower expression in the cortex and hippocampus.” This suggests that the cerebellum should have been a major ROI in this study. Why cerebellum is not included?

5. Figures have extremely low quality (for example Figures 4 and 5). I could only guess the labels and axes.

6. It seems rather strange to have (significant) differences prior to injection (e.g. Fig 5.h)? This needs to be addressed. One would expect to see no difference whatsoever.

7. In favor with reported findings, below paper showed that diffusion MRI is affected by perivascular space fluid presence, which should be cited:

Sepehrband, F., Cabeen, R. P., Choupan, J., Barisano, G., Law, M., Toga, A. W., & Alzheimer's Disease Neuroimaging Initiative. (2019). Perivascular space fluid contributes to diffusion tensor imaging changes in white matter. NeuroImage, 197, 243-254.

8. Not in favor of reported finding, below study reported “we failed to detect a significant change in the brain extracellular water volume using diffusion weighted imaging in awake and anesthetized mice.” This paper also should be cited/discussed:

Gakuba, C., Gaberel, T., Goursaud, S., Bourges, J., Di Palma, C., Quenault, A., ... & Gauberti, M. (2018). General anesthesia inhibits the activity of the “glymphatic system”. Theranostics, 8(3), 710.

9. Another paper that links diffusion changes to glymphatic system is below, which also could be cited:

Thomas, C., Sadeghi, N., Nayak, A., Trefler, A., Sarlls, J., Baker, C. I., & Pierpaoli, C. (2018). Impact of time-of-day on diffusivity measures of brain tissue derived from diffusion tensor imaging. Neuroimage, 173, 25-34.

A minor comment:

1. Page 11, line 251: “peculiar to this this brain” -> “peculiar to this brain”

2. Please report the age of the mice, in each group (maybe it is reported somewhere, but I couldn’t find it).

Reviewer #2: In this paper the dynamic changes of astrocyte activity were investigated using DWI in 32 mice by inhibiting AQP4 channels with a TGN-020 solution. Two novel DWI measures were used to study the results which show a significant decrease in the Sindex, a diffusion marker of tissue microstructure, and a significant increase of the water diffusion coefficient (sADC) in cerebral cortex and hippocampus compared to saline injection.

Developing reliable non-invasive biomarkers for the glymphatic system is important for translational studies. This study has done a great job to introduce such biomarkers. However, it may need some more work to improve the study.

Major points:

1- In addition to sindex, the authors could use FA for comparison or at least confirming (in a few sentences) that FA is not showing the trend seen in sindex.

2- Why mice were chosen? Rats have bigger brain and therefore the imaging could be easier.

3- Histology analyses of the mice after imaging could be so helpful in confirming the study outcomes.

Minor points:

Line 16: “We assumed no change of position among the different scans during acquisitions for each mouse and geometric distortion with b values to be negligible; this condition was qualitatively checked on several mice.” Why not using motion correction to compensate for animal movement during the imaging?

Line 119: “Furthermore, signal instabilities were quantitatively evaluated for each subject and subjects exhibiting instabilities above 4% for most voxels were eliminated.” Could you please explain more?

Line 156: Please mention in Fig3 caption the time point of the displayed maps? I assume the maps correspond to the averaged 6 time points of the pre injection and the last 6 time points after the injection.

Line 251: There is an extra “this”.

6. PLOS authors have the option to publish the peer review history of their article (what does this mean?). If published, this will include your full peer review and any attached files.

Reviewer #1: No

Reviewer #2: No

---

## [Author Response · Author response to Decision Letter 0]

10 Apr 2020

Editor comment

1. How do you know that the observed diffusion changes are related to the glymphatic system?

The existence of the glymphatic system is controversial. Our results do not aim at demonstrating the existence or not of a glymphatic system. However, if we admit that such a system exists it has been strongly suggested in the literature (REF XXX) that astrocytes must play a major role, and, in turn, AQP4 channels expressed by astrocytes. The aim of our work is to show that the modulation of astrocyte activity by TGN-020 which is known as a AQP4 channel blocker can be monitored with diffusion MRI and in particular the Sindex, that’s all. The Discussion and Conclusion have been revised to make this point clear. 

2. How do you explain the DG change, given the mentioned AQP4 abundant?

The density of AQP4 receptors in DG is higher than in the other regions we have investigated. Indeed, the larger sADC and Sindex change observed in DG under TGN-020 administration is an indirect proof that TGN-020 acted on astrocyte AQP4 channels. The Discussion has been revised accordingly.

3. Do you have the data to identify regions with high vascular presence to look at the Si and sADC changes on those area?

A relevant parameter could be, indeed, the density of small vessels. Such vessels are not visible even within the high resolution T2-weighted images we acquired using Turbo RARE (see Methods and Figure 2). Such vessels would require the additional acquisition of perfusion MRI data, such as IVIM MRI, which could not be done during the acquisition time window our anesthetic protocol permitted. Hence, we relied on previous studies reporting the vascular distribution across brain structures using microscopy (ref 40).

4. In addition to sindex, the authors could use FA for comparison.

FA is relevant to assess diffusion anisotropy which is present only in whiter matter at the resolution of our images (not in the cortex). The Sindex (and the sADC) is a much more relevant parameter. Furthermore, as explained in a recent article (Iima at al. European Radiology, 2020), FA values are highly sensitive to noise and, in turn, to ADC (or sADC) values. Variations in FA would thus reflect the changes we have observed in sADC values more than genuine changes in diffusion anisotropy.

Journal Requirements

i) Please amend your current ethics statement to include the full name of the ethics committee that approved your specific study.

ii) Once you have amended this/these statement(s) in the Methods section of the manuscript, please add the same text to the “Ethics Statement” field of the submission form (via “Edit Submission”).

For additional information about PLOS ONE submissions requirements for ethics oversight of animal work, please refer to http://journals.plos.org/plosone/s/submission-guidelines#loc-animal-research

We initially refrained to provide information which could potentially conflict with an anonymous review. Full details have been added in the Methods section.

Response to reviewers

Reviewer #1: Authors presented a very interesting study, evaluating the utility of diffusion MRI to map glymphatic function, through the modification of the AQP4 expression. This is of high scientific and clinical value, very timely and of high impact.

The experiment is well designed, and the execution is sound. This reviewer has below comments regarding the validity and conclusions. Figures quality/dpi are low!

Major comments:

1. The main limitation is interpreting diffusion measures as glymphatic system function. AQP4 is also involved in brain inflammation and regulation of extracellular space volume. How do you know that the observed diffusion changes are related to the glymphatic system?

Even though authors were cautious about the interpretation of their results, they should probably discuss other potential explanations for their findings and adjust their main conclusion accordingly.

The existence of the glymphatic system is controversial. Our results do not aim at demonstrating the existence or not of a glymphatic system. However, if we admit that such a system exists it has been strongly suggested in the literature (Illif et al, Sci Transl Med, 2012,) that astrocytes must play a major role system (Mestre et al, eLife, 2018), and, in turn, AQP4 channels expressed by astrocytes (Ikeshima-Kataoka et al, Int J Mol Sci, 2016). The aim of our work is to show that the modulation of astrocyte activity by TGN-020 which is known as a AQP4 channel blocker can be monitored with diffusion MRI and in particular the Sindex, that’s all. Further studies will be required to establish the potential of diffusion MRI to monitor the glymphatic system. The Discussion and Conclusion have been revised to make this point clear. 

2. Authors reported significant differences in both CA3 and DG of the TGN-020 group. How do you explain the DG change, given the mentioned AQP4 abundant?

The density of AQP4 receptors in DG is higher than in the other regions we have investigated (Hsu et al, Neuroscience, 2011; Hubbard et al, ASN Neuro, 2015). Indeed, the larger sADC and Sindex change observed in DG under TGN-020 administration is an indirect proof that TGN-020 acted on astrocyte AQP4 channels. However, as we explain in the response to comment#6, the baseline of DG was not stable due to the smaller voxel size. We need higher resolution image for the validation of Sindex changes in DG. The Discussion has been revised accordingly.

3. Regarding above comment: Do you have the data (for example SWI or high resolution T2) to identify regions with high vascular presence to look at the S_i and sADC changes on those area? That could potentially help identify the underlying AQP4 involvement.

We agree that a relevant parameter could be, indeed, the density of small vessels. Such vessels are not visible even within the high resolution T2-weighted images we acquired using Turbo RARE (see Methods and Figure 2). Such vessels would require the additional acquisition of perfusion MRI data, such as IVIM MRI, which could not be done during the acquisition time window our anesthetic protocol permitted. Hence, we relied on previous studies reporting the vascular distribution across brain structures using microscopy (Xiong et al, Front Neuroanat, 2017).

4. Based on ref [36]: “AQP4 protein levels were highest in the cerebellum with lower expression in the cortex and hippocampus.” This suggests that the cerebellum should have been a major ROI in this study. Why cerebellum is not included?

We did not have a full brain coverage including the cerebellum due to the limitation of the scanning time (scanning time is determined by number of slices). Instead, we investigated the hippocampus because AQP4 expresses abundantly in this region (Hsu et al, Neuroscience, 2011; Hubbard et al, ASN Neuro, 2015).

5. Figures have extremely low quality (for example Figures 4 and 5). I could only guess the labels and axes.

We are sorry for the very low quality of Figure 4 and 5 which probably resulted from the PDF conversion. Revised figures were uploaded after checking quality with the digital diagnostic tool of PLOS: , https://pacev2.apexcovantage.com/.

6. It seems rather strange to have (significant) differences prior to injection (e.g. Fig 5.h)? This needs to be addressed. One would expect to see no difference whatsoever.

Because the volume of DG is smaller than cerebral cortex and striatum, the Sindex and sADC in DG were noisier rather than these regions. (The number of voxel size in DG, CA3, hippocampus, striatum, cortex is respectively: 85, 181, 763, 2069 and 8270) This is the reason why baseline of Sindex and sADC in DG was not stable. We decided to move DG and CA3 from the manuscript to supplementary data, instead we have used whole hippocampal region for the main figure. We described the instability in DG and CA3 in the Results section. 

7. In favor with reported findings, below paper showed that diffusion MRI is affected by perivascular space fluid presence, which should be cited:

Sepehrband, F., Cabeen, R. P., Choupan, J., Barisano, G., Law, M., Toga, A. W., & Alzheimer's Disease Neuroimaging Initiative. (2019). Perivascular space fluid contributes to diffusion tensor imaging changes in white matter. NeuroImage, 197, 243-254.

We have added this reference in Introduction (Reference 26).

8. Not in favor of reported finding, below study reported “we failed to detect a significant change in the brain extracellular water volume using diffusion weighted imaging in awake and anesthetized mice.” This paper also should be cited/discussed:

Gakuba, C., Gaberel, T., Goursaud, S., Bourges, J., Di Palma, C., Quenault, A., ... & Gauberti, M. (2018). General anesthesia inhibits the activity of the “glymphatic system”. Theranostics, 8(3), 710.

This article compared the ADC and intracerebral accumulation of DOTA-Gd injected in intracisternal site in awaked and anesthetized state. The suppressed accumulation of DOTA-Gd was observed in anesthetized state compared with awaked state, while ADC was not changed between awaked and anesthetized sate. The DOTA-Gd increased the SNR when they accumulated for long time (30-60 min for accumulation in the study) and acquisition has been done under anesthesia. In contrast, diffusion MRI has been performed in awaked and anesthetized condition. The diffusion MRI could be influenced by the head motion and respiration, which could be more significant in awaked state. This potentially interferes ADC measurement to reduce the sensitivity. Further study to improve the sensitivity is required to assess the potential of diffusion MRI to detect the awaked and anesthetized state. We have described this article in Discussion.

9. Another paper that links diffusion changes to glymphatic system is below, which also could be cited:

Thomas, C., Sadeghi, N., Nayak, A., Trefler, A., Sarlls, J., Baker, C. I., & Pierpaoli, C. (2018). Impact of time-of-day on diffusivity measures of brain tissue derived from diffusion tensor imaging. Neuroimage, 173, 25-34.

We thank for these valuable references. We agree that those references would be helpful for the readers to better understand and appreciate our results. This has been cited as reference 27.

A minor comment:

1. Page 11, line 251: “peculiar to this this brain” -> “peculiar to this brain”

We have corrected this typo.

2. Please report the age of the mice, in each group (maybe it is reported somewhere, but I couldn’t find it).

We used the mice aged 4-10 weeks in each group. The age of mice has been added into the Material and Methods part of the manuscript.

Reviewer #2: In this paper the dynamic changes of astrocyte activity were investigated using DWI in 32 mice by inhibiting AQP4 channels with a TGN-020 solution. Two novel DWI measures were used to study the results which show a significant decrease in the Sindex, a diffusion marker of tissue microstructure, and a significant increase of the water diffusion coefficient (sADC) in cerebral cortex and hippocampus compared to saline injection.

Developing reliable non-invasive biomarkers for the glymphatic system is important for translational studies. This study has done a great job to introduce such biomarkers. However, it may need some more work to improve the study.

Major points:

1- In addition to sindex, the authors could use FA for comparison or at least confirming (in a few sentences) that FA is not showing the trend seen in sindex.

FA is relevant to assess diffusion anisotropy which is present only in whiter matter at the resolution of our images. The Sindex (and the sADC) is a much more relevant parameter. Furthermore, as explained in a recent article (Iima at al. European Radiology, 2020), FA values are highly sensitive to noise and, in turn, to ADC (or sADC) values. Variations in FA would thus reflect the changes we have observed in sADC values more than genuine changes in diffusion anisotropy.

2- Why mice were chosen? Rats have bigger brain and therefore the imaging could be easier.

We chose mice instead of rats because our diffusion MRI protocol could be applicable in the future to investigate AQP-4 KO mice. There is no such transgenic model for rats. Also, our MRI setup includes a cryoprobe dedicated to mouse head imaging, which increases the SNR and contrast. This specific coil is too small for rat heads. 

3- Histology analyses of the mice after imaging could be so helpful in confirming the study outcomes.

We agree that histology would have been a nice addition to confirm volume changes of astrocytes in the different groups (TGN-020 group and vehicle group). However, such tests require the brain to be fixed, hence they cannot be performed to check volume changes before and after TGN-020 (or saline) injection within the same animal. Another method to monitor astrocyte volume could be in vivo fluorescent imaging, but this method can be applied only in the cortex. Those issues have been added to the Discussion.

Minor points:

Line 16: “We assumed no change of position among the different scans during acquisitions for each mouse and geometric distortion with b values to be negligible; this condition was qualitatively checked on several mice.” Why not using motion correction to compensate for animal movement during the imaging?

Motion correction is tricky when considering quantitative diffusion MRI as motion algorithms work on the reconstructed images. Especially, as acquisitions are performed with specific orientations of the diffusion-probing gradient pulses in space any rotation in the reconstructed images would not reflect genuine signal changes which would have occurred if the diffusion-probing gradient pulses had rotated in the same way. Instead, we decided to simply reject data sets were motion was too large (>4%) using a semi-quantitative index based on signal stability across successive scans. We agree that the text was not clear and we have revised it.

Line 119: “Furthermore, signal instabilities were quantitatively evaluated for each subject and subjects exhibiting instabilities above 4% for most voxels were eliminated.” Could you please explain more?

The standard-deviation of the signal intensity across scans was evaluated for each b value and each voxel (after Gaussian smoothing) was estimated. A threshold of 4% was used flag unstable voxels. . Data sets were one of the ROIs have only flagged voxels were discarded. 

Line 156: Please mention in Fig3 caption the time point of the displayed maps? I assume the maps correspond to the averaged 6 time points of the pre injection and the last 6 time points after the injection.

We have added in the legend of the figure 3 that the maps correspond to the mean of the 6 last time points.

Line 251: There is an extra “this”.

We have corrected this typo.

---

## [Decision Letter · Decision Letter 1]

12 Apr 2020

PONE-D-20-03675R1

Diffusion MRI reveals in vivo and non-invasively changes in astrocyte function induced by an aquaporin-4 inhibitor.

PLOS ONE

Dear Pr. Le Bihan,

Thank you for submitting your manuscript to PLOS ONE. After careful consideration, we feel that it has merit but does not fully meet PLOS ONE’s publication criteria as it currently stands. Therefore, we invite you to submit a revised version of the manuscript that addresses the points raised during the review process.

Please response reviewer's minor question.

We would appreciate receiving your revised manuscript by May 27 2020 11:59PM. To enhance the reproducibility of your results, we recommend that if applicable you deposit your laboratory protocols in protocols.io, where a protocol can be assigned its own identifier (DOI) such that it can be cited independently in the future. For instructions see: http://journals.plos.org/plosone/s/submission-guidelines#loc-laboratory-protocols

We look forward to receiving your revised manuscript.

Kind regards,

Quan Jiang, Ph,D.

Academic Editor

PLOS ONE

Reviewers' comments:

Reviewer's Responses to Questions

**Comments to the Author**

1. If the authors have adequately addressed your comments raised in a previous round of review and you feel that this manuscript is now acceptable for publication, you may indicate that here to bypass the “Comments to the Author” section, enter your conflict of interest statement in the “Confidential to Editor” section, and submit your "Accept" recommendation.

Reviewer #1: (No Response)

Reviewer #2: All comments have been addressed

2. Is the manuscript technically sound, and do the data support the conclusions?

Reviewer #1: Yes

Reviewer #2: Yes

3. Has the statistical analysis been performed appropriately and rigorously? 

Reviewer #1: I Don't Know

Reviewer #2: Yes

4. Have the authors made all data underlying the findings in their manuscript fully available?

Reviewer #1: No

Reviewer #2: Yes

5. Is the manuscript presented in an intelligible fashion and written in standard English?

Reviewer #1: Yes

Reviewer #2: Yes

6. Review Comments to the Author

Reviewer #1: Authors responded to the raised issues and addressed reviewer’s concern.

One point that remained unaddressed is the general assumption. The author states that “The aim of our work is to show that the modulation of astrocyte activity by TGN-020 which is known as a AQP4 channel blocker can be monitored with diffusion MRI and in particular the Sindex, that’s all.”

But throughout the paper the text suggests that that’s not all. For example, the abstract says

“The Glymphatic System (GS) has been proposed as a mechanism to clear brain tissue from waste. Its dysfunction might lead to several brain pathologies, including the Alzheimer’disease. A key component of the GS and brain tissue water circulation is the astrocyte which is regulated by acquaporin-4 (AQP4), a membrane-bound water channel on the astrocytic end-feet.”

Well, clearly the author is relating this to glymphatic system. Yet, relating this to glymphatic system is not the main concern. If one can measure astrocyte activity, it’s not unreasonable to relate it to glymphatic system. The main concern is that AQP-4 is not only involved in modulation of astrocyte activity. It is also involved in the brain inflammatory response and also in the regulation of the extracellular volume. Both of these could affect diffusion signal. Blocking AQP-4 could lead to inflammation or changes in the extracellular fluid which affect the diffusion signal. Therefore, what you observe here may have nothing to do with the astrocyte activity. This limitation should be addressed.

PS. Regarding DG, if the data is noisy, it should lead to higher standard deviation not a systematic group mean difference.

Reviewer #2: The authors resolved well my concerns about their work and made the manuscript more clear. I have no other question to add.

7. PLOS authors have the option to publish the peer review history of their article (what does this mean?). If published, this will include your full peer review and any attached files.

Reviewer #1: No

Reviewer #2: No

---

## [Author Response · Author response to Decision Letter 1]

28 Apr 2020

Reviewer #1: Authors responded to the raised issues and addressed reviewer’s concern.

One point that remained unaddressed is the general assumption. The author states that “The aim of our work is to show that the modulation of astrocyte activity by TGN-020 which is known as a AQP4 channel blocker can be monitored with diffusion MRI and in particular the Sindex, that’s all.”

But throughout the paper the text suggests that that’s not all. For example, the abstract says

“The Glymphatic System (GS) has been proposed as a mechanism to clear brain tissue from waste. Its dysfunction might lead to several brain pathologies, including the Alzheimer’disease. A key component of the GS and brain tissue water circulation is the astrocyte which is regulated by acquaporin-4 (AQP4), a membrane-bound water channel on the astrocytic end-feet.”

Well, clearly the author is relating this to glymphatic system. Yet, relating this to glymphatic system is not the main concern. If one can measure astrocyte activity, it’s not unreasonable to relate it to glymphatic system. The main concern is that AQP-4 is not only involved in modulation of astrocyte activity. It is also involved in the brain inflammatory response and also in the regulation of the extracellular volume. Both of these could affect diffusion signal. Blocking AQP-4 could lead to inflammation or changes in the extracellular fluid which affect the diffusion signal. Therefore, what you observe here may have nothing to do with the astrocyte activity. This limitation should be addressed.

PS. Regarding DG, if the data is noisy, it should lead to higher standard deviation not a systematic group mean difference

The statement about the Glymphatic System in the abstract and the introduction is only to explain our general motivation to conduct our study of the potential of diffusion MRI to monitor astrocyte activity. It is not a claim that the GS exists (we believe we have been careful in the manuscript about this fact). We only briefly review the literature to shade light on the context of the study and indicate why investigating astrocyte activity might be important. We are aware that TGN-020 effects are not specific to astrocytes, as stated in the manuscript. Effects on the brain inflammatory response can be discarded given the time course of the effects we have observed. Regarding the extracellular space (ISF) effects on the diffusion MRI would result only from changes in diffusion hindrance/tortuosity (fluid flow is well too slow to result in IVIM effects, this has been shown by other groups, furthermore the Sindex was purposely calculated using the lowest key b value 250s/mm² to make residual IVIM effects completely negligible). Hence, variations in extracellular space shape and volume only reflect changes in the local cellular background (a kind of negative imprint). The decrease in Sindex (and sADC increase) points out to decrease in the amount of diffusion hindrance from astrocytes (which is by far the dominant cell type in the regions where we observed the largest response). As stated in the discussion “Based on established diffusion MRI mechanisms [28] this hindrance decrease suggests an astrocyte volume reduction [43,44] associated with an increase of the ISF (were diffusion is tortuous) [45] or an increase in astrocyte membrane permeability and water exchange”. We believe that this statement about the mechanism leading to our observation is correct and broad enough, and the manuscript was not changed.

Regarding the DG, the data are, indeed, very noisy given the very small voxel count. The apparent group difference may just be coincidental as the standard-deviation cannot be accurately estimated precisely given the small voxel count. We realize that this might be puzzling, but, for scientific integrity, we decided not to “hide” those results but to show them. A note has been added to the caption of SuppL Fig.1.

---

## [Decision Letter · Decision Letter 2]

29 Apr 2020

Diffusion MRI reveals in vivo and non-invasively changes in astrocyte function induced by an aquaporin-4 inhibitor.

PONE-D-20-03675R2

Dear Dr. Le Bihan,

We are pleased to inform you that your manuscript has been judged scientifically suitable for publication and will be formally accepted for publication once it complies with all outstanding technical requirements.

With kind regards,

Quan Jiang, Ph,D.

Academic Editor

PLOS ONE

Additional Editor Comments (optional):

Reviewers' comments:

Reviewer's Responses to Questions

**Comments to the Author**

1. If the authors have adequately addressed your comments raised in a previous round of review and you feel that this manuscript is now acceptable for publication, you may indicate that here to bypass the “Comments to the Author” section, enter your conflict of interest statement in the “Confidential to Editor” section, and submit your "Accept" recommendation.

Reviewer #1: All comments have been addressed

2. Is the manuscript technically sound, and do the data support the conclusions?

Reviewer #1: Yes

3. Has the statistical analysis been performed appropriately and rigorously? 

Reviewer #1: Yes

4. Have the authors made all data underlying the findings in their manuscript fully available?

Reviewer #1: No

5. Is the manuscript presented in an intelligible fashion and written in standard English?

Reviewer #1: Yes

6. Review Comments to the Author

Reviewer #1: This reviewer was not convinced by authors respond, claiming extracellular changes will not affect Sindex. However this does not damper reviewers enthusiasm about this interesting work.

I congratulate the authors and wish them well.

7. PLOS authors have the option to publish the peer review history of their article (what does this mean?). If published, this will include your full peer review and any attached files.

Reviewer #1: No

---

## [Editor Report · Acceptance letter]

4 May 2020

PONE-D-20-03675R2 

Diffusion MRI reveals in vivo and non-invasively changes in astrocyte function induced by an aquaporin-4 inhibitor. 

Dear Dr. Le Bihan:

I am pleased to inform you that your manuscript has been deemed suitable for publication in PLOS ONE. Congratulations! Your manuscript is now with our production department. 

With kind regards,

on behalf of

Dr. Quan Jiang 

Academic Editor

PLOS ONE